# AI-ming backwards: Vanishing archaeological landscapes in Mesopotamia and automatic detection of sites on CORONA imagery

Alessandro Pistola[1]*, Valentina Orrù[2]*, Nicolò Marchetti[2‡], Marco Roccetti[1‡]*

**1** Department of Computer Science and Engineering, University of Bologna, Bologna, Italy, **2** Department of History and Cultures, University of Bologna, Italy

☯ These authors contributed equally to this work.
‡ These authors also contributed equally to this work.
* marco.roccetti@unibo.it

## Abstract

By upgrading an existing deep learning model with the knowledge provided by one of the oldest sets of grayscale satellite imagery, known as CORONA, we improved the AI model's attitude towards the automatic identification of archaeological sites in an environment which has been completely transformed in the last five decades, including the complete destruction of many of those same sites. The initial Bing-based convolutional network model was re-trained using CORONA satellite imagery for the district of Abu Ghraib, west of Baghdad, central Mesopotamian floodplain. The results were twofold and surprising. First, the detection precision obtained on the area of interest increased sensibly: in particular, the *Intersection-over-Union* (IoU) values, at the image segmentation level, surpassed 85%, while the general accuracy in detecting archeological sites reached 90%. Second, our re-trained model allowed the identification of four new sites of archaeological interest (confirmed through field verification), previously not identified by archaeologists with traditional techniques. This has confirmed the efficacy of using AI techniques and the CORONA imagery from the 1960s to discover archaeological sites currently no longer visible, a concrete breakthrough with significant consequences for the study of landscapes with vanishing archaeological evidence induced by anthropization.

## Introduction

For the study of archaeological landscapes in the Near East and the reconstruction of settlement patterns therein, the primary objective is the identification, localization and chronological characterization of ancient settlements: those considered in this analysis are predominantly tells, both because they represent the most widespread type of ancient settlement evidence in the region under study and because of their high visibility starting already from preliminary remote sensing approaches. A tell is

**Data availability statement:** All results were obtained using open-source software and models, as well as publicly available data (images, annotations) and computational resources (Google Colab), making this type of work highly accessible and replicable even in resource-limited research environments. In addition to the specific information provided within the document, all the code, data, archeological annotations and various resources are available on GitHub (https://github.com/alepistola/AI_floodplains).

**Funding:** The 2022-2024 Abu Ghraib survey project, directed by N. Marchetti, has been authorized by the Iraqi State Board of Antiquities and Heritage (SBAH) and funded by the University of Bologna and the Italian Ministry of Foreign Affairs and International Cooperation. The research presented in this paper was supported through the "KALAM. Analysis, protection and development of archaeological landscapes in Iraq and Uzbekistan through ICTs and community-based approaches" project, coordinated by N. Marchetti and funded by the Volkswagen Foundation (www.kalam.unibo.it). The funders had no role in study design, data collection and analysis, decision to publish, or preparation of the manuscript.

**Competing interests:** The authors have declared that no competing interests exist.

an artificial mound created by the accumulated debris from centuries of human habitation. These mounds typically form in regions such as the Mesopotamian floodplain, where communities repeatedly built and rebuilt their settlements at the same site with perishable material. Prior to any field verification, the very first step for site identification is remote sensing, which has been a key resource for archaeologists for many years, being a non-invasive method that contributes to the detection and preservation of cultural heritage, it requires, however, a large amount of expert human work and consequently requires a significant amount of time [1–6]. It goes without saying that using deep learning techniques as a support to human efforts could open new perspectives. For example, deep learning abilities can be put to good use for automatically analyzing satellite imagery, especially using a technique called *semantic segmentation* which, in simple words, consists of assigning a class label to each pixel in an image, up to a point where the entire image is recognized as easily interpretable by archaeologists. Several works in this field have confirmed the efficacy of this approach [7–10]. Finally, it is worth mentioning that since the time when we began our research activity in the area of Deep Learning applied to archaeology, many other similar initiatives have taken shape and have been brought to the forefront of scientific discussion. We are well aware that a wealth of studies have made relevant advances in this field [11–14]. In order to avoid any misunderstanding, we should state at the onset that here are (at least) two axes along which specialists can look at similarities or differences in these kinds of research. The first one concerns the remote sensing methodologies employed to gather the data on which various Artificial Intelligence techniques can work. Using passive or active approaches, satellites or aircrafts, radar or LiDAR or any other typology of sensors emitting their own signals, or a combination of some of them, constitutes a relevant difference that extends till the point of a distinction (or divergence) in the meaning of the results which can be obtained, irrespective of the type of data, since working with new rather than old (e.g., CORONA, in our specific case) high resolution imagery represents another important source of differentiation. The second axis is that of the distinction between Machine Learning (ML) and Deep Learning (DL). ML are algorithms that learn from *structured* data to predict outputs and discover patterns in that data. DL, instead, is always based on highly complex neural networks that mimic the way a human brain works to detect patterns in large *unstructured* data (like images). A traditional ML algorithm can be something as simple as linear regression or a search in a decisional tree, the driving force behind being often that of ordinary statistics. DL algorithms, instead, should be regarded as a sophisticated and mathematically complex evolution of ML. To achieve this result, DL mechanisms use a layered structure of algorithms, called artificial neural networks, with a specific design based on a cascade of several different computational blocks, inspired by the biological neural network of the human brain, and leading to a process of learning that is far more capable than that of standard ML models. It should be also considered that with DL one can often fall into an excess of inference to which it is difficult (even not possible) to give a formal explanation. An extension of this discussion, tailored to the archaeological field, has been reported elsewhere [15]. Consequently, applying either ML or DL makes

an important difference, being often unfair (or nonsensical), in the light of the explanation above, a comparison between studies that adopt different approaches. Given these premises, we look at the wealth of researches that have investigated how well AI techniques can work in the archaeological field, not with the aim of conducting one-to-one comparisons with specific papers that could have followed different approaches, but rather with the responsibility of witnessing the level of productivity of the entire community in this specific field.

## Our previous work

Building on a long-term scientific collaboration between AI-ers and archaeologists at the University of Bologna, Italy [16–19], a deep learning model was recently proposed, enhanced with segmentation and self-attention mechanisms, which was able to detect mounded archaeological sites in the Mesopotamian floodplain in southern Iraq. A set of modern (Bing-based) georeferenced vector shapes were used as data source, corresponding to the outlines of the previously mentioned *tells* and surrounding areas, totaling 4934 shapefiles in the southern Mesopotamian floodplain, which constitute the entirety of the surveyed and published sites in the area [Floodplains Project; 15]. Each image in that dataset was subjected to a variety of image manipulation techniques (including, for example, random rotation, mirroring, brightness and contrast correction) and it was then given as an input to train a convolutional neural network, augmented with segmentation and self-attention mechanisms. In the end, the result of this activity was a deep learning model able to detect archaeological sites in the area of interest, which achieved during a test on a set of already known sites an *Intersection Over Union* (IoU) score of 0.81, with a general accuracy in the neighborhood of 80%. This first work had, nonetheless, two important limitations. First, an initial attempt to exploit CORONA satellite imagery was unsuccessful, probably due to our inability to integrate this panchromatic imagery with full color pictures. Second, the entire testing activity was conducted on hundreds of already known archaeological sites, without the possibility of challenging the machine predictions on sites not already groundtruthed. Computer scientists' wish, instead, would have been to make those automatic predictions on sites not already certified as tells and, upon confirmation by the archaeologists, subjecting them to a process of ground-truthing, to understand in reality whether those predictions were accurate or not.

## New developments

The study area selected for this research is the district of Abu Ghraib, located in central Iraq within the Baghdad Governorate. This region, which had never been the subject of systematic archaeological investigations, except for its easternmost edge explored by Robert McCormick Adams [20], was also chosen due to the observed high degree of landscape transformation over recent decades. This contribution is part of a broader landscape archaeology project grounded in well-established traditional methodologies, including conventional remote sensing [21,22], field validation with collection and study of associated archaeological evidence, spatial analysis, integrated with AI-based approaches. Between 2023 and 2024, a systematic field survey was carried out with the aim of verifying on the ground all potential sites first identified through remote sensing, both by conventional analysis and by AI-based predictive models. The field investigations confirmed the presence or absence of archaeological remains and allowed, when necessary, for the refinement of site boundaries.

Since the inception of the project, a central role was played by the assessment of threats to the archaeological heritage. This analysis, a key component of the traditional methodological framework, was conducted by comparing recent satellite imagery with historical CORONA images acquired between 1960 and 1972 through the U.S. reconnaissance program [23–28]. The remote sensing analysis systematically documented landscape transformations, identifying major causes of site damage such as agricultural expansion, canal digging, and urban encroachment. These preliminary findings were subsequently validated and further investigated through fieldwork, which allowed for a more precise assessment of the current condition of each site. The results revealed that 38% of sites had been completely destroyed,

23% had lost more than half of their original extent, and only 38% retained more than half of their surface area, as it will be detailed in the final Section of this manuscript. As a direct consequence of these findings, a methodological decision was made to focus the training of AI-based predictive models on CORONA imagery, as the only source capable of capturing the archaeological landscape before extensive modern transformations. In essence, the study presented here thus focuses on the use of CORONA imagery to improve the performance of AI-based predictive models. Our previous convolutional neural network was retrained using transfer learning techniques and subjected to a two-stage fine-tuning process, resulting in three distinct configurations: (1) one based exclusively on Bing imagery, (2) one using only CORONA imagery, and (3) a combined configuration. The validation of results was carried out in two phases: the first on known sites and areas without archaeological evidence, and the second on new predictions generated by the models, which were also verified through fieldwork. The findings demonstrate that the integration of historical imagery significantly enhances the ability of AI models to detect archaeological sites, confirming the effectiveness of an approach that combines historical sources, technological innovation, and traditional archaeological methods in the reconstruction of historical Mesopotamian landscapes.

## Materials and methods

We first describe the data used in our study, and then we illustrate the methods employed to build our AI models (for accessing all the developed software and the data used in this study, see the Section: Data Availability Statement).

### Data

We begin by noticing that all the remote sensing operations we carried out to identify archaeological evidence and to extract the usable data were conducted on the basis of various publicly available basemaps, specifically: current (Google, Bing, and Esri) and historical satellite imagery (CORONA), and topographic maps (US Army 1:100,000 from 1942). During this phase, 88 potential tells were identified, recorded as vector shapefiles and classified with the abbreviation *GHR* (i.e., the initials of the geographical district of interest: Abu Ghraib) and a sequential number. Starting from that information, the image creation process was based on the following five steps: **i)** all the shapefiles of the area of interest were imported from Bing and CORONA basemaps into an open-source GIS software [QGIS; [29]], **ii)** sample squares each 2000 meters long, centered on the centroid of any given shapefile, were extracted from those images using a Python script developed by us (this was done in the same way both for Bing, through the QuickMapService plugin, and for the CORONA imagery, via the free services provided by the University of Arkansas' Center for Advanced Spatial Technologies). At that point, **iii)** we generated the truth masks, that is the masks that put in evidence, at a pixel level, the points either included in a tell or not. After that phase, we were in the obvious situation of having an unbalanced dataset, with a prevalence of non-empty truth masks. To fill this gap, additional images, with an empty truth mask, were generated, to balance the dataset. This was done, **iv)** by choosing 120 random points in the area of interest (with relative surrounding images, not containing any tell). The final dataset consisted of 88 images (around 41%) each including a *tell*, and 120 images (almost 59%) not portraying any tell or its parts, totaling a final amount of 208 pictures, on which a training activity could be conducted. Nonetheless, given the relatively small size of this dataset, **v)** an *aggressive* data augmentation procedure was exploited that has prevented the overfitting phenomenon. In particular, using the Albumentations library [30], three subsequent transformations (geometric, color space and kernel filters) were applied to all the images (Bing, CORONA and truth masks), where each transformation was chosen, in turn, from one of the three separate groups shown in Table 1, with a given probability. Fig 1 provides three examples of such a pipelined image augmentation process, where the transformations (RandomCrop, Flip, RandomRotate90, GaussNoise, Sharpen, Resize), (RandomCrop, Flip, RandomRotate90, CLAHE, GaussNoise, Sharpen, Resize) and (RandomCrop, Flip, RandomRotate90, RandomBrightnessContrast, MotionBlur, Sharpen, Resize) were applied in the reported cases following that exact sequence.

**Table 1. Data augmentation pipeline.**

| Group | Technique | Probability of use |
|---|---|---|
| | RandomCrop | 1 |
| | Flip | 0.5 |
| | RandomRotate90 | 0.5 |
| Geometric | | 0.2 |
| | GridDistorsion | 0.4 |
| | RandomGridShuffle | 0.6 |
| Color space | | 0.5 |
| | CLAHE | 0.4 |
| | RandomBrightnessContrast | 0.8 |
| | ChannelShuffle | 0.1 |
| | ColorJitter | 0.2 |
| | HueSaturationValue | 0.2 |
| Kernel filters | | 0.5 |
| | Blur | 0.4 |
| | GaussNoise | 0.4 |
| | MotionBlur | 0.2 |
| | Sharpen | 0.1 |
| | Resize | 1 |

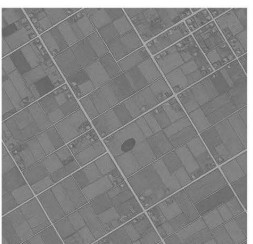 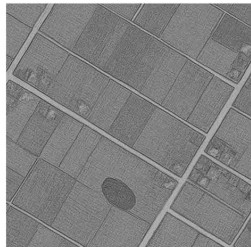 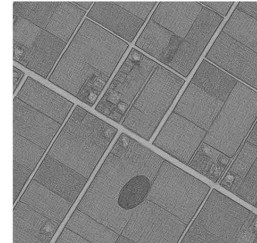 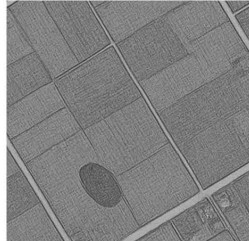

**Fig 1. Image augmentation pipeline: an example. Disclaimer**: All the displayed data fall under the condition of fair use utilization of geographical data for academic purposes. The list of all relevant data/software provider(s) is as follows: (i) satellite imagery is based on Copernicus Sentinel-2 data, freely available under the European Union's open data policy (https://www.copernicus.eu/en/access-data/copernicus-open-access-hub); (ii) maps display achieved with open source software, under the GNU licenses of QGIS (https://qgis.org/en/site/) and QuickMapServices (https://github.com/nextgis/quickmapservices); (iii) final maps elaboration achieved with a software developed by the authors and available at (https://github.com/alepistola/AI_floodplains).

## Methods

The type of convolutional neural network and the method used to train it were similar to those adopted in our previous works [18,19]. Nevertheless, while a detailed description may be found there, for the benefit of readers one may remind as follows: our work started by using the PyTorch Segmentation Models library and by defining a Deep Convolutional Neural Network (DCNN) model, called MANet. Its complex architecture is summarized in Fig 2. A MANet (Multi-scale Attention Network) is a deep-learning neural network tailored to learn robust and discriminative features from high resolution (remote sensing) images. It stacks various multi-scale attention blocks, aiming at reducing spatial and channel redundancy to accelerate convolution. As shown in Fig 2, it is composed of three main blocks that act in sequence, one after another: an encoder, a decoder and a segmentation head. The encoder, represented by the topmost block in the central box of Fig 2,

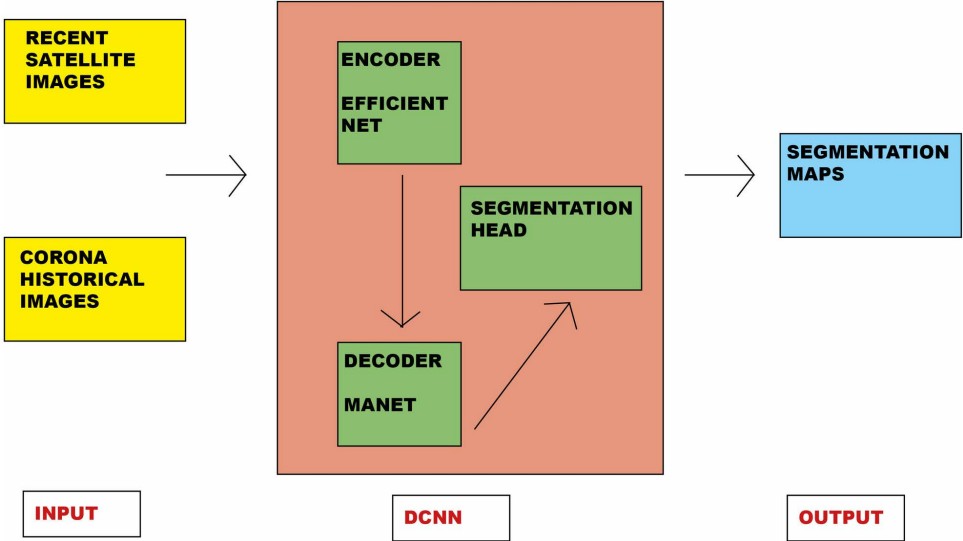

**Fig 2. Stylized diagram of the reference architecture for the DCNN with Attention used in this study (with input and output). Disclaimer**: This figure was completely drawn by the authors in every part and is an entirely original product of their ingenuity. It does not utilize any graphical elements (in whole or in part) belonging to others, and consequently, it is in no way dependent on terms of use dictated by others. The authors, as creators of the Figure, authorize its publication for academic purposes and grant its use under a CC BY license, with attribution to them.

constitutes a proper convolutional architecture, based on the well-known Efficientnet-b3 model [31,32]. Input to this encoder are high resolution images (leftmost box of Fig 2), with a given number of channels (as satellite sensors can collect images at various regions of the electromagnetic spectrum) and a corresponding spatial resolution expressed as a matrix of pixels (n x n). This imagery passes, within the encoder, through a cascade of multiple sub-blocks, which implement a convolution procedure, followed by subsequent operations of batch normalization and swish activation. Essentially, the actual image convolution procedure takes place inside the encoder, which consists in converting many pixels within their receptive field (the area of the input that influences a particular feature) into single values, aiming at reducing the number of free parameters, while allowing the network to be deeper. In the end, in our case, the final layers of the encoder return feature maps at a reduced resolution (16 x16) over 384 channels. With a capillary flow of information from the encoder to the decoder, the latter is activated. The decoder represents the true operational core of a MANet (the bottommost block in the central box of Fig 2). Its role is to perform a weighted recombination of the features extracted by the encoder, with the ultimate goal of returning segmentation maps (i.e., segmentation shapes). Those segmentation maps are the most-informative output of a MANet, as they can be used to highlight the sub-regions of interest (archaeological in our case) within the original input images. Our convolutional network, as previously mentioned, implements attention mechanisms. To this end, two important sub-blocks of the decoder are dedicated, specifically a Position-wise Attention Block (or PAB) and a sequence of Multi-scale Fusion Attention Blocks (or MFABs). Regarding these attention mechanisms, they allow our models to weigh various latent features within the images, effectively directing the model's attention in this latent space for improved learning [32]. More precisely, the PAB incorporates a positional encoding mechanism. This mechanism produces an attention map, which effectively pinpoints pixels of greater significance, guiding our architecture to identify regions that require segmentation. Furthermore, through the cascading arrangement of several MFABs, we have implemented a multi-scale strategy. This method aggregates features to capture inter-channel relationships, thereby enhancing the robustness of the segmentation. Finally, we have the block at the rightmost position in the central box of Fig 2, which represents the segmentation head. This is the final layer, responsible for computing the ultimate segmentation maps. It is the last step where the initial remote

sensing images are partitioned into distinct regions, with their pixels homogeneously classified to identify sharper boundaries. At this point, however, as previously mentioned, the segmentation shapes are still at a low resolution. This is where the up-sampling blocks come in, simply used to return output maps (rightmost box of Fig 2) with the same resolution as the input images. In closing, it is worth noticing that, at the completion of its function, the entire procedure is able to process around ten high resolution images in less than a second.

Starting from the DCNN architecture discussed above, we have re-trained our models, initially pre-trained on both Imagenet and on the images exploited in our previous work, with the new 208 images introduced in the previous Section, resorting to traditional transfer learning techniques [19,33,34]. In essence, we aimed at obtaining three new deep learning models, that added to the three ones built during our previous study [19], where the re-training activity was based on the use of the new imagery, respectively, provided by Bing, CORONA and a combination of both. As already anticipated, after these training activities, we concluded with a further incremental step of fine-tuning, applied to all the AI models of interest, using a particular technique called two-stage fine-tuning [35]. Technically speaking, this additional two-stage fine tuning activity, included, in turn, a first phase where, keeping the learning rate unchanged, the weights of the deep layers were frozen, subjecting to training only the *segmentation head*. In the second phase of this procedure, instead, the weights were unfrozen, reducing the learning rate by a factor of ten, and carrying out a re-training of the entire model, thus reducing the risk of both overfitting and catastrophic forgetting. We conducted this final procedure with the number of training epochs not fixed and proceeded until we detected a stagnation of the loss on the validation set, indicating a possible overfitting to avoid. For the sake of clarity, we summarize this (only apparently) complex situation providing, in the list below, all the six different models, differentiated based on the combination of the training activities to which they were subjected and the type of image dataset used:

• **Bing**: the deep learning model trained on Bing basemaps during our previous study [19].

• **Bing_Bing**: the deep learning model previously trained on Bing basemaps and now re-trained on new Bing basemaps and finally fine-tuned as explained below.

• **CORONA**: the deep learning model trained on CORONA basemaps during our previous study [19].

• **CORONA_CORONA**: the deep learning model previously trained on CORONA basemaps and now re-trained on new CORONA basemaps and finally fine-tuned as explained below.

• **BingCORONA**: the deep learning model trained on a combination of Bing and CORONA basemaps during our previous study [19].

• **BingCORONA_BingCORONA**: the deep learning model previously trained on a combination of Bing and CORONA basemaps and now re-trained on new Bing and CORONA basemaps and finally fine-tuned as explained below.

Moving to the issue of the type of metrics used to evaluate the efficacy of our system, it is worth mentioning that the accuracy returned by our models was evaluated on the basis of the consideration of two different assessment perspectives: that is, a) through the lens of the semantic segmentation to which each image was subjected (i.e., trying to evaluate the achieved accuracy only at a pixel level), and b) at a more general level, analyzing the confusion matrix, to obtain an assessment of the accuracy and recall values. In particular, to evaluate the results produced by segmentation, we have used three different metrics: i) the Intersection over Union (IoU), ii) the binary Intersection Over Union, (bIoU), and finally iii) the Matthews Correlation Coefficient (MCC). The mathematical definitions of these metrics are beyond the scope of this paper and can be easily retrieved from the specialized literature, more interesting are the motivations for this choice which are as follows. The IoU metric is largely used in similar situations, but it may present various defects as it is recognized that it can return high values that could be not directly related to a better general object recognition, but just to a more precise identification of its contours. Indeed, we used it in this study, because it allowed a comparison with previous

results. Its variant, the bIoU, is instead a better candidate to measure the accuracy of detection in terms of the pixels being recognized as a part of a tell. Finally, the measurement of MCC values was added as it should represent the most appropriate metric for the problem under consideration based on findings described in recent literature [36,37]. Coming to the final phase of our study, upon assessment of the performances of our AI models we chose the one with better accuracy: its results were plotted under the form of a corresponding heatmap, and then passed to the archaeological team on the field. Based on these heatmaps, the latter made the final decision on which were the more promising sites deserving a visit during the field survey campaigns.

## Results

We must preliminarily mention that, of the 208 initial images, only 10% (i.e., 20 images) were used as subjects of the testing phase. In fact, 156 images (75%) were used for re-training our models, with 32 of them (15%) used during the validation phase. Before discussing the results achieved on the 20 images which were never shown to our models before this testing phase, we deem of interest to report also on the results obtained with the 32 images of the validation phase. Nonetheless, it should be clear that this phase (i.e., the validation phase) constitutes an integral part of the re-training activities discussed in the previous Section. As such, the corresponding results do not represent the ultimate measurement of how well our models recognize tells when new imagery is proposed. Rather, they have given a preliminary assurance that our models have learnt effectively during re-training, with a generic propensity to generalize well to new images. Obviously, monitoring this tendency during training helped us to fine-tune our models for better results. With this in mind, Table 2 provides the results obtained with the 32 images of the validation phase. The following factors should be taken into consideration. First, being validation a part of the re-training activity, these results are provided only for the three re-trained models, namely: Bing_Bing, Bing_CORONA_BingCORONA, and CORONA_CORONA. Second, the numerical values of Table 2 are not given under the form of average and standard deviation, as they represent, instead, the better values the models achieved (during a specific epoch) before overfitting occurred. Third, these results only focus on the pixel-wise accuracy.

Table 3 reports, instead, the average results (plus standard deviation) we achieved with our testing activity conducted with the 20 images our models have never seen before. These results are based on the metrics that we have already indicated being the most appropriate to recognize a tell at a pixel level (that is: IoU, bIoU and MCC). Each test was repeated ten times.

**Table 2. Validation: pixel-wise accuracy for the three re-trained models.**

| Model | IoU | MCC | bIoU | Epoch |
|---|---|---|---|---|
| **Bing_Bing** | 83.07 | 55.25 | 37.00 | 15 |
| **Bing_CORONA_BingCORONA** | 84.42 | 69.99 | 56.86 | 9 |
| **CORONA_CORONA** | 82.25 | 59.30 | 43.70 | 27 |

**Table 3. Testing: pixel-wise accuracy for all models (average values with standard deviation).**

| Model | IoU | St.d. | MCC | St.d. | bIoU | St.d. |
|---|---|---|---|---|---|---|
| Bing (previous) | 82.24 | 2.88 | 35.24 | 6.36 | 22.70 | 5.55 |
| **Bing_Bing** | **86.12** | 2.77 | **34.03** | 9.95 | **21.53** | 8.67 |
| Bing_CORONA (previous) | 84.30 | 1.56 | 45.76 | 7.93 | 28.80 | 6.45 |
| **Bing_CORONA_BingCORONA** | **85.77** | 2.03 | **55.63** | 5.88 | **39.23** | 6.37 |
| CORONA (previous) | 83.54 | 2.02 | 31.98 | 8.57 | 18.80 | 5.91 |
| **CORONA_CORONA** | **85.09** | 3.32 | **47.27** | 9.84 | **33.19** | 8.84 |

The results of Table 3 show that the re-training activity we conducted in the present study, combined with the effects of the two-stage fine tuning procedure, has had very positive effects on both the so called **BingCORONA_BingCORONA** and **CORONA_CORONA** models, also when compared with the results of our previous study, yielding a notable increase in terms of all the considered metrics. As to the **BING_BING** model, instead, it only improves on the IoU parameter. The following fact is of great interest: as the most notable increase in accuracy has been achieved in both models based on CORONA satellite imagery, while the simple Bing model presented no significant variation, this adds experimental evidence to the intuition that the combination of the two stage fine tuning procedure with the activities of transfer learning becomes really effective (with more accurate results) only when the corresponding model was built on top of the **CORONA** imagery. While it is true that in other researches, including the one we conducted previously, the integration of CORONA satellite imagery into generic AI models had produced inconclusive results (with motivations ranging from low resolution up to environmental factors, like cloud cover for example), the present study supports the hypothesis that the inclusion of CORONA imagery has the potential to enhance an AI model's performance that has the task of recognizing *tells* from satellite imagery. In other words, the improvement we have measured, at a pixel level, substantiates the thesis that transfer learning and complex fine-tuning activities may benefit from the additional contextual information provided by these kinds of imagery, thus corroborating long-established archaeological insights of the same sign. Nonetheless, the results of Table 3 have (simply) informed us about the ability of our models to recognize if a given pixel is either comprised within a tell or not. We are, obviously, also interested in elevating our comprehension about the ability of our AI models to recognize a tell as a whole. Table 4 gives an answer to this question by providing the results we achieved in terms of the general accuracy in detecting tells, as emerging from the testing activities conducted with the same 20 images mentioned before. The used metrics, here, were those of *accuracy* and *recall* (the mathematical definitions of which can be easily retrieved from the specialized literature), while TP stands for true positives, TN: true negatives, FP: false positives and FN: false negatives. Table 3, again, highlights the increased ability of the BingCORONA_BingCORONA model in detecting tells, reaching a detection accuracy in the neighborhood of 90% (while our previous results hardly surpassed 80% [19]), with a very low percentage of both false positives and negatives.

### New discoveries

Beyond the positive results reported in Tables 2, 3, and 4, the novelty of our work lies in the idea to use machine predictions (upon approval of the domain experts) to decide to extend the set of archaeological sites to be inspected during field survey campaigns, which we did. As already anticipated, our best AI model (i.e., BingCORONA_BingCORONA) produced prediction heatmaps, like that shown in Fig 3.

These heatmaps were analyzed by the archaeologists who compared them with the list of sites of potential interest identified through standard remote sensing operations. In our case, of which the heatmap in Fig 3 is an example, our attention was mainly attracted by machine predictions for a number of specific sites that were not previously recognized as potential tells before through traditional methods. Of these sites, eight were accompanied with very high values of probability of being positive cases as returned by the AI model, which led to one of the key results presented in this paper. Subsequently, two field reconnaissance campaigns were conducted in January 2023 and January 2024, covering the Iraqi district of Abu Ghraib at the northwestern apex of the Mesopotamian floodplain. Field activities were directed at verifying

**Table 4. Testing: tell detection (accuracy and recall).**

| Model | Accuracy | Recall | TP | TN | FP | FN |
|---|---|---|---|---|---|---|
| Bing_Bing | 0.75 | 0.50 | 4 | 11 | 1 | 4 |
| BingCORONA_BingCORONA | **0.90** | **0.88** | 7 | 11 | 1 | 1 |
| CORONA_CORONA | 0.70 | 0.67 | 4 | 10 | 4 | 2 |

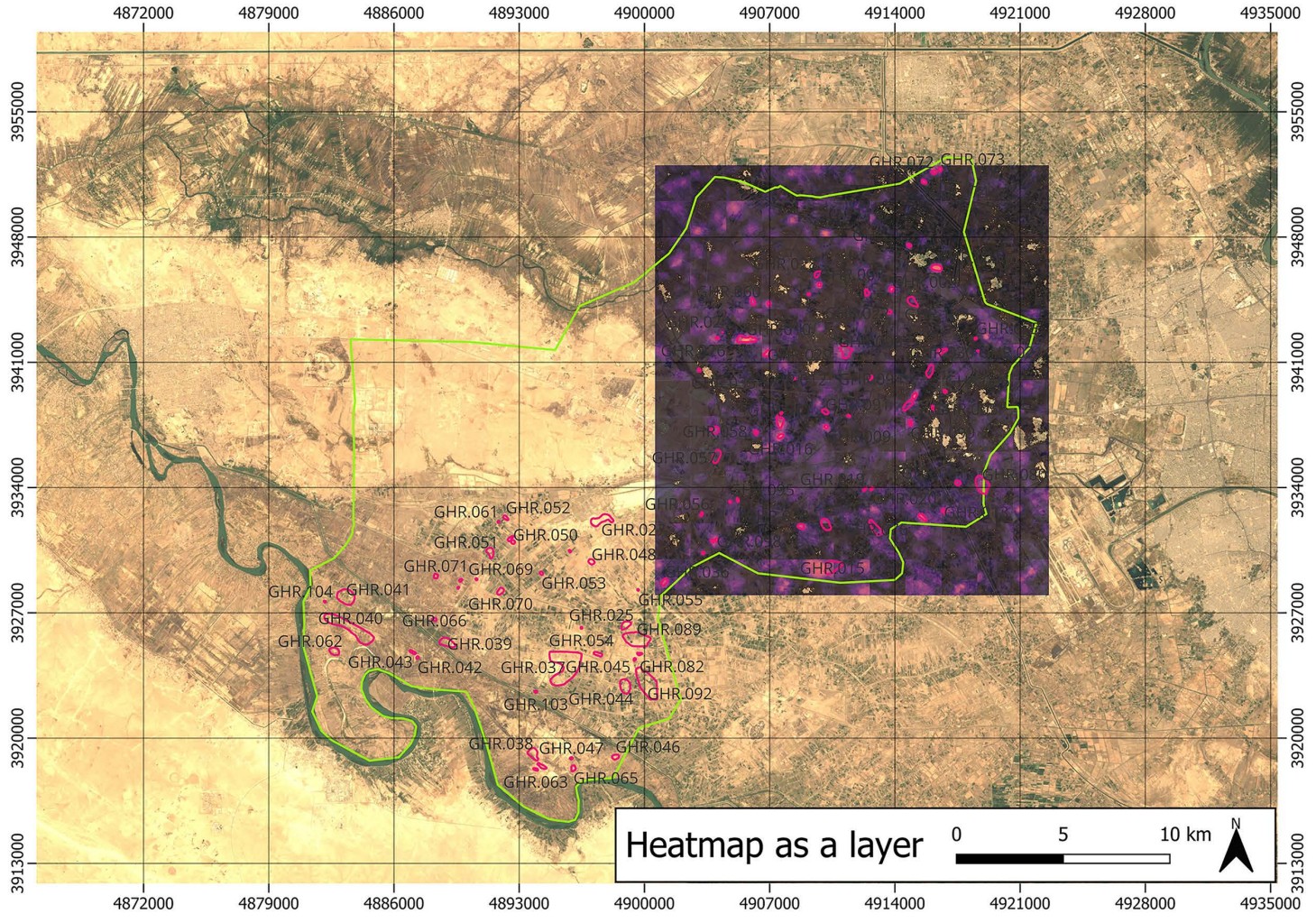

**Fig 3. Example of an AI-generated heatmap used to predict the presence of archaeological sites. Disclaimer**: All the displayed data fall under the condition of fair use utilization of geographical data for academic purposes. The list of all relevant data/software provider(s) is as follows: (i) satellite imagery is based on Copernicus Sentinel-2 data, freely available under the European Union's open data policy (https://www.copernicus.eu/en/access-data/copernicus-open-access-hub); (ii) maps display achieved with open source software, under the GNU licenses of QGIS (https://qgis.org/en/site/) and QuickMapServices (https://github.com/nextgis/quickmapservices); (iii) final maps elaboration achieved with a software developed by the authors and available at (https://github.com/alepistola/AI_floodplains).

sites identified using the CORONA imagery (both those identified with standard remote sensing procedures and those suggested by the AI model described above). During these two campaigns, a total of 96 potential sites were investigated (including the eight suggested by the AI). Of these 96, only 15 showed no signs of ancient anthropogenic activity and were thus false positives. The field survey results revealed, in fact, that 81 turned out to be positively confirmed sites. Of the eight sites suggested by the AI, four were among the 81 confirmed sites of archaeological relevance. To be noticed, again, is the fact that all the 81 sites were discovered by virtue of the analyses conducted on the CORONA satellite imagery, being based either on remote sensing or through AI. The validation of these sites was achieved through the collection and subsequent study of superficial ancient ceramics, which also enabled their dating. Fig 4 summarizes these field-survey results, showing the entire survey area inside which both remote sensing- and AI- based predicted sites are shown using dots of different colors (also based on the fact they were confirmed as either positive or negative cases).

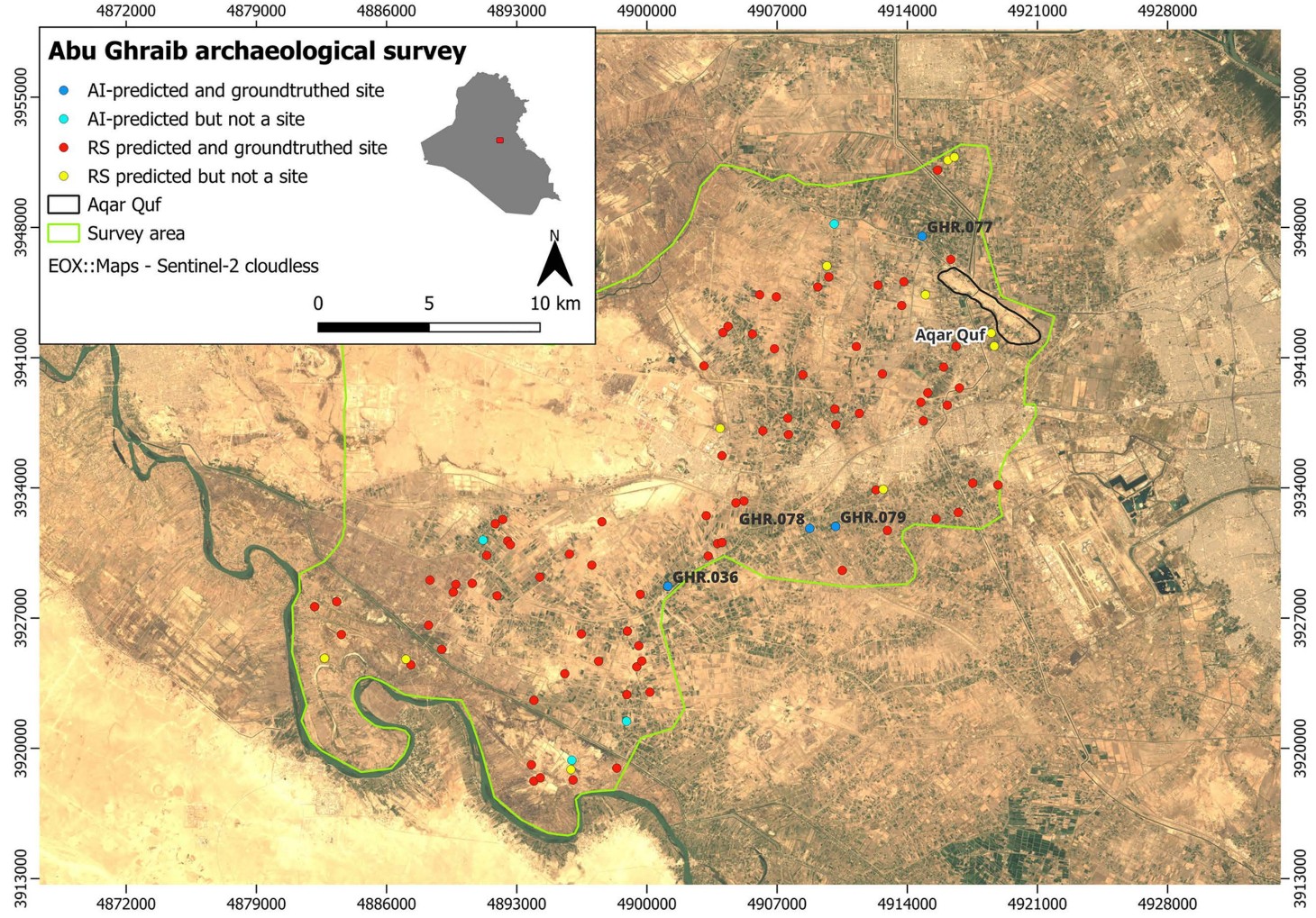

**Fig 4. Sites discovered during the Abu Ghraib archaeological survey campaigns. Red: positive cases discovered by AI. Blue: positive cases discovered with remote sensing. Disclaimer**: All the displayed data fall under the condition of fair use utilization of geographical data for academic purposes. The list of all relevant data/software provider(s) is as follows: (i) satellite imagery is based on Copernicus Sentinel-2 data, freely available under the European Union's open data policy (https://www.copernicus.eu/en/access-data/copernicus-open-access-hub); (ii) maps display achieved with open source software, under the GNU licenses of QGIS (https://qgis.org/en/site/) and QuickMapServices (https://github.com/nextgis/quickmapservices); (iii) final maps elaboration achieved with a software developed by the authors and available at (https://github.com/alepistola/AI_floodplains).

As to the four archaeological sites discovered based on the suggestion of our BingCORONA_BingCORONA deep learning model, Figs 5–8 show, to the left, the heatmap produced with the CORONA imagery and, to the right, an actual ground photo of the site (all the geographical coordinates of these four confirmed sites are listed in the S1 Appendix below).

## Discussion

In this research, the use of AI techniques has been of great help in support to a process which remains, nonetheless, guided by the archaeologist's knowledge and expertise. Deep learning models have helped towards the aim to identify areas potentially containing archaeological sites, albeit neglected during normal remote sensing operations. It has remained an archaeologists' task to take the final decision about the precise locations of the sites to visit, based on

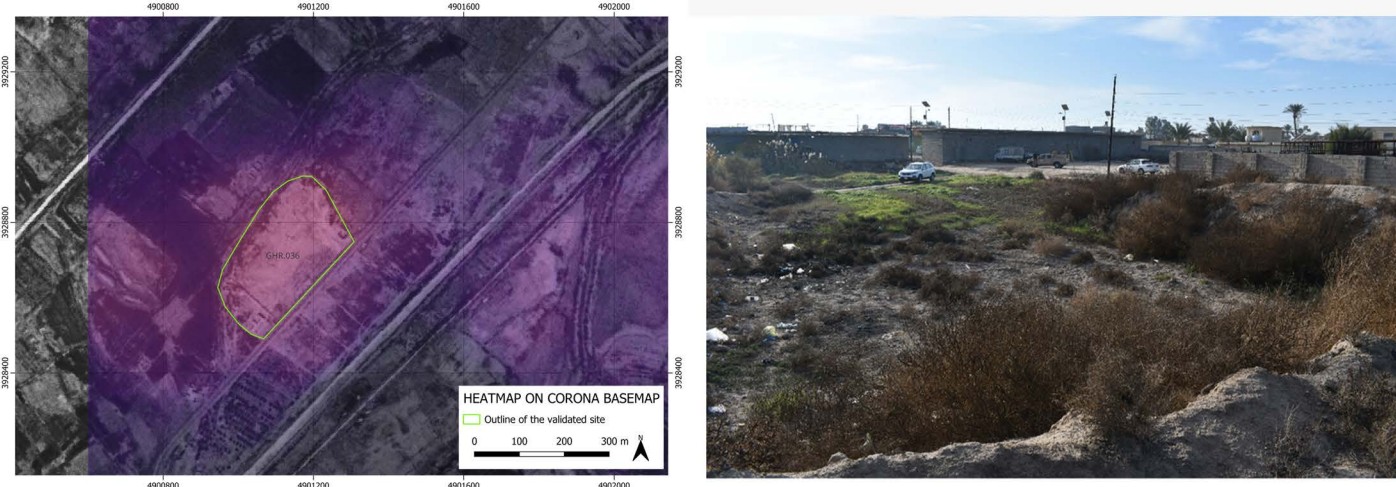

**Fig 5. GHR.036: heatmap produced by our model on CORONA imagery (left) and an on-site photo overview (right). Disclaimer**: All the displayed data fall under the condition of fair use utilization of geographical data for academic purposes. The list of all relevant data/software provider(s) is as follows: (i) CORONA satellite imagery utilized here is freely available through the United States Geological Survey (USGS) under its Open Data Policy (https://www.usgs.gov/faqs/are-usgs-reportspublications-copyrighted); (ii) maps display achieved with open source software, under the GNU licenses of QGIS (https://qgis.org/en/site/) and QuickMapServices (https://github.com/nextgis/quickmapservices); (iii) final maps elaboration achieved with a software developed by the authors and available at (https://github.com/alepistola/AI_floodplains).

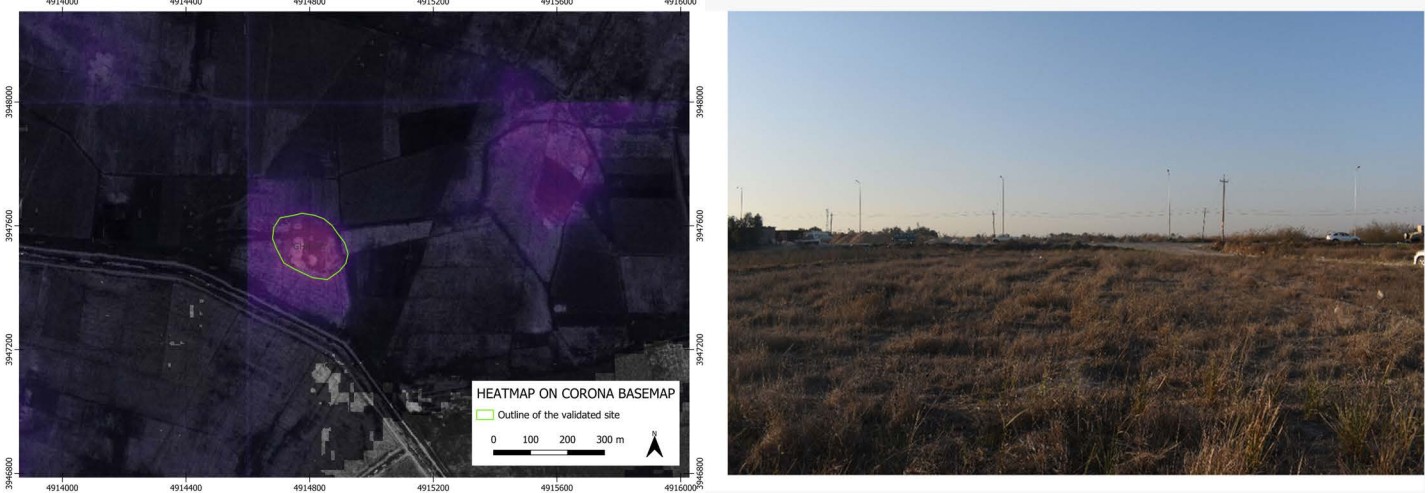

**Fig 6. GHR.077: heatmap produced by our model on CORONA imagery (left) and an on-site photo overview (right). Disclaimer**: All the displayed data fall under the condition of fair use utilization of geographical data for academic purposes. The list of all relevant data/software provider(s) is as follows: (i) CORONA satellite imagery utilized here is freely available through the United States Geological Survey (USGS) under its Open Data Policy (https://www.usgs.gov/faqs/are-usgs-reportspublications-copyrighted); (ii) maps display achieved with open source software, under the GNU licenses of QGIS (https://qgis.org/en/site/) and QuickMapServices (https://github.com/nextgis/quickmapservices); (iii) final maps elaboration achieved with a software developed by the authors and available at (https://github.com/alepistola/AI_floodplains).

their professional experience. In this sense, our proposed AI-based approach to archaeology has been conceived just to provide additional support to the archaeologists, rather than to replace them. Beyond the impact of the AI, in some sense already documented in a previous study of ours [19], it has emerged here also the fundamental role played by

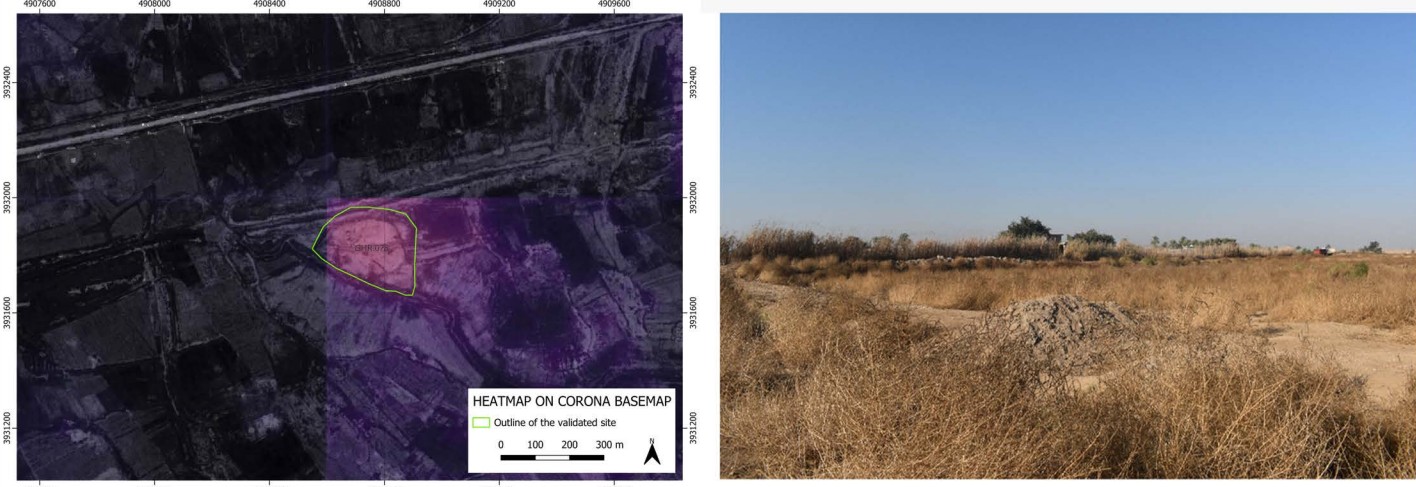

**Fig 7. GHR.078: heatmap produced by our model on CORONA imagery (left) and an on-site photo overview (right).** **Disclaimer**: All the displayed data fall under the condition of fair use utilization of geographical data for academic purposes. The list of all relevant data/software provider(s) is as follows: (i) CORONA satellite imagery utilized here is freely available through the United States Geological Survey (USGS) under its Open Data Policy (https://www.usgs.gov/faqs/are-usgs-reportspublications-copyrighted); (ii) maps display achieved with open source software, under the GNU licenses of QGIS (https://qgis.org/en/site/) and QuickMapServices (https://github.com/nextgis/quickmapservices); (iii) final maps elaboration achieved with a software developed by the authors and available at (https://github.com/alepistola/AI_floodplains).

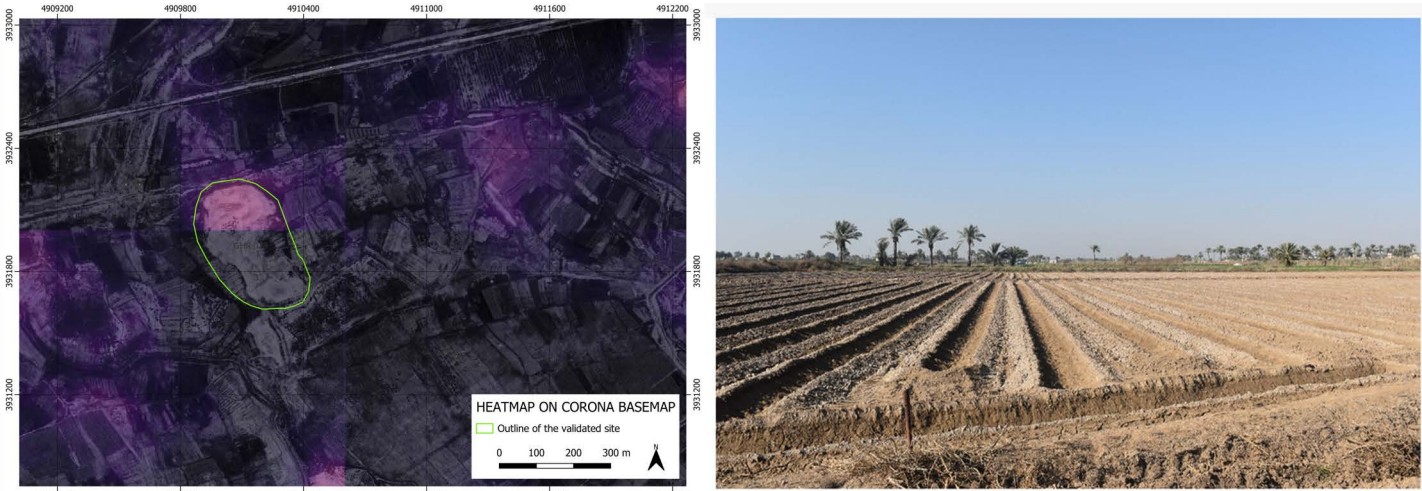

**Fig 8. GHR.079: heatmap produced by our model on CORONA imagery (left) and an on-site photo overview (right).** **Disclaimer**: All the displayed data fall under the condition of fair use utilization of geographical data for academic purposes. The list of all relevant data/software provider(s) is as follows: (i) CORONA satellite imagery utilized here is freely available through the United States Geological Survey (USGS) under its Open Data Policy (https://www.usgs.gov/faqs/are-usgs-reportspublications-copyrighted); (ii) maps display achieved with open source software, under the GNU licenses of QGIS (https://qgis.org/en/site/) and QuickMapServices (https://github.com/nextgis/quickmapservices); (iii) final maps elaboration achieved with a software developed by the authors and available at (https://github.com/alepistola/AI_floodplains).

the CORONA imagery dataset, and its versatility, especially when used in combination with a deep learning model. This consideration derives not only from our direct experience on the four archaeological sites which were not detectable using the Bing imagery alone, thereby highlighting the substantive impact of incorporating the CORONA satellite imagery into

the process, but also considering the state of preservation of the sites which were the subject of the on-field campaigns of 2023 and 2024. To better illustrate this point, Fig 9 shows that, of the 81 archaeological sites discovered during those campaigns, almost all had been destroyed over the past decades: either completely (31) or largely (19) or partially (31); where *completely* means a destruction of almost 100%, *largely* means over the threshold of 50% and finally *partially* means below 50%. In this context, the inclusion of CORONA satellite imagery has been fundamental because many of the destroyed sites were no longer visible from modern basemaps (like Bing maps). The CORONA satellite imagery, from the 1960s and early 1970s, has the ability to document a world that has almost disappeared: in the specific case of Abu Ghraib, the loss of the possibility to identify sites with modern basemaps, in fact, would range from 40% to 55%, if totally destroyed sites alone, or totally plus largely destroyed ones, were considered. Thus, the development of an automatic

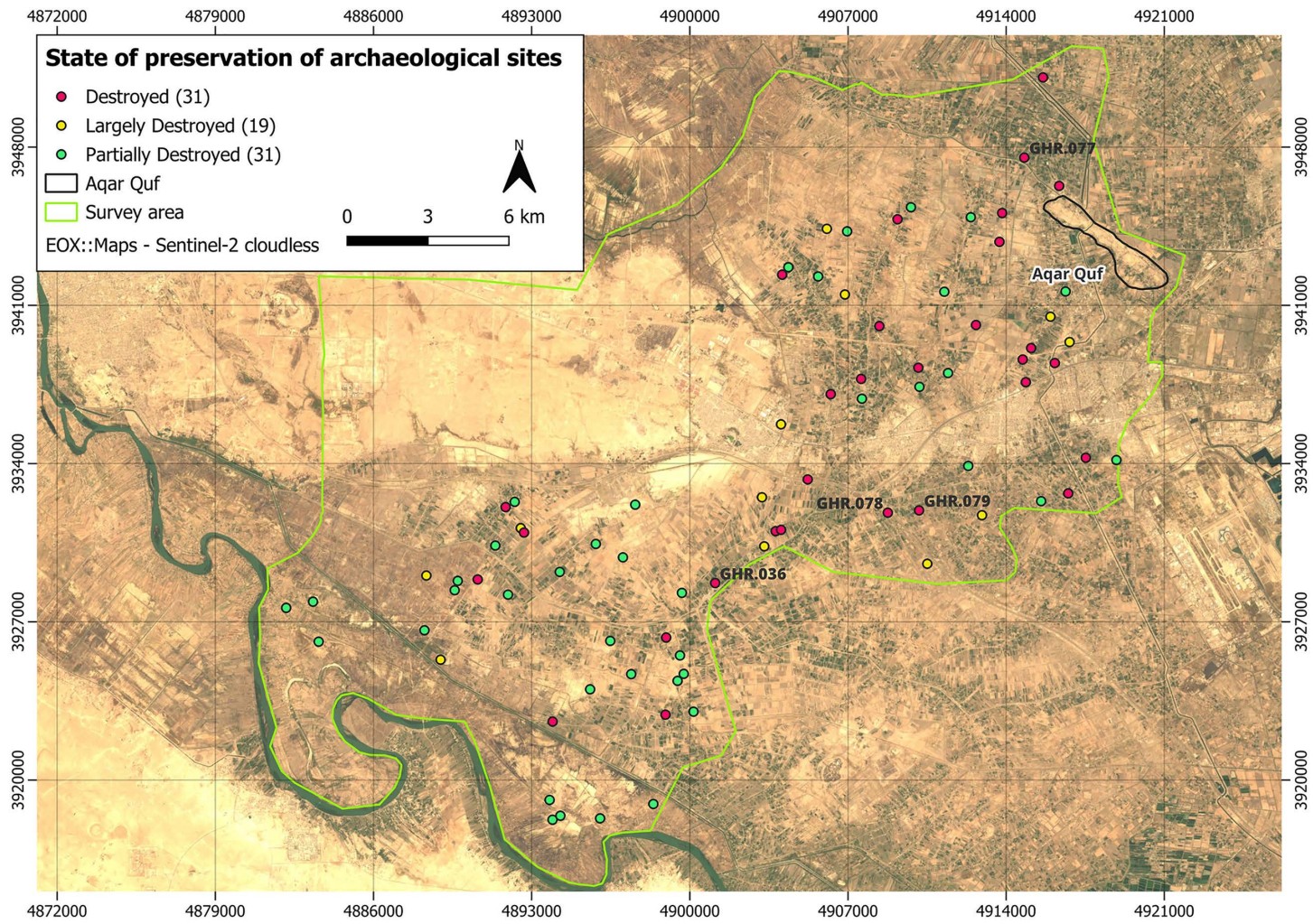

**Fig 9. State of preservation of the 81 archaeological sites discovered during the 2023-2024 on-field campaigns.** **Disclaimer**: All the displayed data fall under the condition of fair use utilization of geographical data for academic purposes. The list of all relevant data/software provider(s) is as follows: (i) satellite imagery is based on Copernicus Sentinel-2 data, freely available under the European Union's open data policy (https://www.copernicus.eu/en/access-data/copernicus-open-access-hub); (ii) maps display achieved with open source software, under the GNU licenses of QGIS (https://qgis.org/en/site/) and QuickMapServices (https://github.com/nextgis/quickmapservices); (iii) final maps elaboration achieved with a software developed by the authors and available at (https://github.com/alepistola/AI_floodplains).

process that is able to identify disappearing sites, by including historical imagery, allows everyone to start a fundamental reflection for the protection of the existing/remaining archaeological evidence.

To conclude this Section, we would like to add that, while we have documented that an AI-based identification process has the potential to make unexpected discoveries, nonetheless, what should not be forgotten is the awareness that we still do not know how this happens. Precisely, this should be the reason that pushes towards the integration of AI with human experts, through collaborative processes, aimed at mitigating classification errors and incorrect interpretations [38–42].

## Conclusion

We have described a deep learning model designed to identify sites of potential archaeological interest in the Abu Ghraib district, West of Baghdad in the Mesopotamian floodplain. This AI model has been built incrementally over the past years, using transfer learning techniques and a final two stage fine tuning procedure that has elevated the level of detection accuracy up to 90% (while previous results did not surpass the threshold of 80%). The role played by the CORONA imagery dataset has been fundamental in this context of vanishing archaeological evidence, as it has allowed the AI to *see* sites no longer visible due to the process of anthropization. Surprisingly, this process has also led to the identification of unexpected archaeological sites, which thus far had not been identified in standard remote sensing operations. In particular, our archaeological team visited the eight new sites suggested by the AI model, also because they had the appropriate morphological characteristics. During our field survey campaign, four of these eight sites have been confirmed as positive cases. In fact, even if they were totally destroyed and no longer visible on the ground, some ceramic sherds could still be collected, making it possible to confirm their existence and date them. It must be acknowledged that without the AI's suggestions, the areas where the sites were confirmed would not have been investigated during routinary field surveys. In the end, the development of AI models able to automatically identify potential sites, no more visible from current basemaps, represents a real breakthrough which could be further extended exploring the possibility of adding other technologies and methods like, for example, LIDAR and super-resolution ones [43–49]. While our work has focused on tell-based sites—characterized by distinctive morphologies well-suited to automatic segmentation—it is worth noting that extending this approach to non-mounded contexts would represent a theoretically significant development. However, the lack of recurrent morphological features, the semantic heterogeneity of archaeological traces, and the current scarcity of annotated datasets make such a direction presently difficult to pursue. Advancing in this area would require fundamentally different classification strategies, substantial refinement of source data, and a methodological rethinking that goes beyond the aims and operational scope of the present study.

## Supporting information

**S1 Appendix. Geographical coordinates of GHR.036, GHR.077, GHR.078 and GHR.079 archaeological sites.** (DOCX)

## Author contributions

**Conceptualization:** Nicolò Marchetti, Marco Roccetti.

**Data curation:** Alessandro Pistola, Valentina Orrù.

**Formal analysis:** Alessandro Pistola.

**Funding acquisition:** Nicolò Marchetti.

**Investigation:** Valentina Orrù, Nicolò Marchetti.

**Methodology:** Nicolò Marchetti, Marco Roccetti.

**Software:** Alessandro Pistola.

**Supervision:** Nicolò Marchetti, Marco Roccetti.

**Visualization:** Alessandro Pistola, Valentina Orrù.

**Writing – original draft:** Marco Roccetti.

**Writing – review & editing:** Alessandro Pistola, Valentina Orrù, Nicolò Marchetti, Marco Roccetti.

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
