## [Decision Letter · Decision Letter 0]

16 Mar 2025

Dear Dr. ROCCETTI,

Thank you for submitting your manuscript to PLOS ONE. After careful consideration, we feel that it has merit but does not fully meet PLOS ONE’s publication criteria as it currently stands. Therefore, we invite you to submit a revised version of the manuscript that addresses the points raised during the review process.

We look forward to receiving your revised manuscript.

Kind regards,

Shai Gordin, Ph.D.

Academic Editor

PLOS ONE

Journal Requirements:

2. In your manuscript, please provide additional information regarding the specimens used in your study. Ensure that you have reported human remain specimen numbers and complete repository information, including museum name and geographic location.

For more information on PLOS ONE's requirements for paleontology and archeology research, see https://journals.plos.org/plosone/s/submission-guidelines#loc-paleontology-and-archaeology-research .

“The 2022-2024 Abu Ghraib survey project, directed by N. Marchetti, has been authorized by the Iraqi State Board of Antiquities and Heritage (SBAH) and funded by the University of Bologna and the Italian Ministry of Foreign Affairs and International Cooperation. The research presented in this paper was supported through the “KALAM”. Analysis, protection and development of archaeological landscapes in Iraq and Uzbekistan through ICTs and community-based approaches” project, coordinated by N. Marchetti and funded by the Volkswagen Foundation (www.kalam.unibo.it).”

Reviewers' comments:

Reviewer's Responses to Questions

**Comments to the Author**

1. Is the manuscript technically sound, and do the data support the conclusions?

Reviewer #1: Partly

Reviewer #2: No

2. Has the statistical analysis been performed appropriately and rigorously?

Reviewer #1: Yes

Reviewer #2: No

3. Have the authors made all data underlying the findings in their manuscript fully available?

Reviewer #1: Yes

Reviewer #2: Yes

4. Is the manuscript presented in an intelligible fashion and written in standard English?

Reviewer #1: Yes

Reviewer #2: No

Reviewer #1: This is a paper focused on using deep learning techniques to auto-discover archaeological sites in Mesopotamia. The paper reflects a growing trend in the use of AI-based techniques to automate discovery of archaeological sites. Overall, the paper is interesting but requires more work prior to publication. I list some of my comments below.

1). The review of previous literature misses quite a lot of developments in the area of remote sensing and archaeology using AI-based techniques. There have been various studies on also using mounds on maps or imagery to find sites. Although many of these do not focus on CORONA, perhaps incorporating these could be useful as it shows where the methodology is best understood in this growing literature. I think incorporating what has been done with UAVs, non-optical data, and other areas could be useful for better understanding many of these related techniques and ultimately how this work is different.

2) The methodology, which is described and presented, needs a lot more description in the main text in my opinion. I think a better discussion on the steps and algorithm is critical. I understand that the details are best presented elsewhere, such as the code, but the paper should also have enough detail to give the reader they understand the mechanics of the technique. To me this needs more work and detail. There is not much discussion on the loss function, parameterisation, and other steps.

3). Its good to see the tests done regarding accuracy. However, do we get a sense of sites being missed by survey and this technique? I know it is hard to tell without knowing what sites exist but random survey of the region studied could help identify what is being missed by these more focused approaches that are looking at areas seen on the results.

4). Although tell sites are obviously common, many sites are flat in Mesopotamia or show little mounding. I understand that these might be difficult on optical data but this may mean other data should be used such as LiDAR (e.g., from UAVs), multispectral, and even higher resolution optical. It seems to me many sites could be missed with this technique. In fact, if one were to use Digital Elevation Data and use that to find sites, as tells become obvious with that data, could we not simply use that to find mounded sites? I would have tested how accurate this technique is relative to someone using DEM data to find sites and see which is more accurate. In any case, I think more discussion on the limitations of the research and why other data sources could not be used to find non-tell (mounded) sites is warranted as there are now a number of papers on finding mounded sites using deep learning in archaeology (see my comment on the literature review).

5. More discussion on usability of this would be good. For instance, on scaling the approach, time it takes to do this work, and benefits of this technique versus other techniques in the literature, such as Mask R-CNN, Random Forest, and other DL techniques.

Reviewer #2: Overall, this is an interesting article employing a methodology which I hope to become more developed and common in broadscale settlement analysis. However, a concern is that the authors present this method as a novel approach, when uses of machine learning algorithms have been employed to this end before in similar areas (Menze et al 2006, Soroush et al 2020, Guyot et al 2018, Orengo et al 2020, etc.). It is concerning that other studies such as these are not cited. I am also concerned at the apparent conflation of “tells” with archaeological sites more broadly. One of the main benefits of this technology is that it should not just be able to identify tells, which are easily identifiable without ML. However, throughout the article, it is difficult to understand if the authors are referring only to the identification of tells (which seems to be the case) or all archaeological sites. Certainly, there are plenty of “flat” sites.

My second main concern with this article is that the AI/ML approach used to identify sites does not actually assist the preexisting survey methods in a meaningful way. If I am reading this correctly, then it appears that the ultimate result of this study is 4 sites and 4 false positives identified that were not initially discovered during survey. However, when compared to the 80+ sites that were identified and surveyed, this number seems almost negligible and not at all like a “real breakthrough” that the authors purport it to be. In fact, in the conclusions section they suggest that these "discovered" sites by the model had "the appropriate morphological characteristics" - does that mean that normal exploratory foot survey would have discovered these sites? Ultimately, it doesn't seem that the model made much of a difference at all to the results of a survey that was using CORONA imagery to identify potential sites in the first place. The authors have given information for the accuracy of the different models they tested, but how does it compare to the accuracy of "normal" site identification with satellite imagery? While the accuracy here is 50/50 in terms of "newly identified" sites, in my experience a normal positive-false positive ratio is more akin to 60-75%-40-25%. (It is also unclear why CORONA makes such a difference in the case of this model - I'm assuming it's the result of modern urbanization and industrial agriculture which have now obscured these sites, but this is unclear in the text itself.)

I think that an article discussing this approach has merit, but should be tested on a larger dataset with more distinct results to merit publication in this journal. My conclusion is to reject this manuscript, but with hope that the authors will continue to develop and apply this model to further material to produce meaningful results in the future.

**Do you want your identity to be public for this peer review?** For information about this choice, including consent withdrawal, please see our Privacy Policy

Reviewer #1: No

Reviewer #2: No

---

## [Author Response · Author response to Decision Letter 1]

23 Apr 2025

We attached a letter for Reviewers as requested.

---

## [Decision Letter · Decision Letter 1]

27 Jun 2025

Dear Dr. ROCCETTI,

Thank you for submitting your manuscript to PLOS ONE. After careful consideration, we feel that it has merit but does not fully meet PLOS ONE’s publication criteria as it currently stands. Therefore, we invite you to submit a revised version of the manuscript that addresses the points raised during the review process.

We look forward to receiving your revised manuscript.

Kind regards,

Shai Gordin, Ph.D.

Academic Editor

PLOS ONE

Journal Requirements:

Reviewers' comments:

Reviewer's Responses to Questions

**Comments to the Author**

Reviewer #1: (No Response)

2. Is the manuscript technically sound, and do the data support the conclusions?

Reviewer #1: Partly

3. Has the statistical analysis been performed appropriately and rigorously?

Reviewer #1: No

4. Have the authors made all data underlying the findings in their manuscript fully available?

Reviewer #1: Yes

5. Is the manuscript presented in an intelligible fashion and written in standard English?

Reviewer #1: Yes

Reviewer #1: I still find a few things missing in this work based on this current version. I think they need to be addressed before the manuscript can be accepted. I list these below.

1) The review of existing literature in this area is still somewhat minimal. I know this is not a review paper, so referencing all works is not that critical but I think more around the specific architecture chosen and the background into that would help because there are other architectures one could have chose and it is not clear to me why this particular method say is better or more appropriate over other CNN-based or DL models/architectures.

2) Validation test are somewhat limited. A broader range of statistical tests could be applied to test how accurate and reliable the quality of the image identification is. Usually I would expect a wider range, including RMSE and other tests to be there.

3) I still think more can be done with the non-tell based sites but I can accept to leave that for now. Some discussion at least as to how this could be an improvement in the future might be warranted (i.e., how to improve non-tell based site identification as this would really be transformative if we can go beyound mound identification).

**Do you want your identity to be public for this peer review?** For information about this choice, including consent withdrawal, please see our Privacy Policy

Reviewer #1: No

---

## [Author Response · Author response to Decision Letter 2]

23 Jul 2025

Dear Editor,

as suggested to us, we have revised our paper: “AI-ming backwards: Vanishing archaeological landscapes in Mesopotamia and automatic detection of sites on CORONA imagery”, submitted for a possible publication to PLOS ONE.

We have taken into serious consideration ALL the comments and suggestions that the Reviewer has made, and we have modified our paper accordingly (further adding one reference and one new discussion).

A letter answering to the comments of the Reviewer is reported below.

It describes how we have modified (or not) our paper in response to all his/her comments. The modifications we have made, have been highlighted with the reddish color in the text of the revised manuscript.

Sincerely,

The Authors

------

Answers to the Reviewer’s Comments:

1) The review of existing literature in this area is still somewhat minimal. I know this is not a review paper, so referencing all works is not that critical but I think more around the specific architecture chosen and the background into that would help because there are other architectures one could have chose and it is not clear to me why this particular method say is better or more appropriate over other CNN-based or DL models/architectures

Answer to 1)

We feel confident in asserting that a list of 48 references, with more than half addressing issues related to AI, allied technologies, and their application in the archaeological domain, is already sufficiently broad and encompassing as to preclude the need for further expansion.

Nonetheless, solely to provide a further confirmation regarding the type of impact our work on the use of AI techniques on satellite data has had in the scientific world, even outside the archaeological context, we are adding a very recent reference, number 49 below.

In this reference, experts from the Chinese government, from one of the most prestigious Chinese laboratories (located in Wuhan), engage with our method, aiming to use satellite data for calculating plausible routes for orbiting towards the planet Jupiter.

[49] Afzal Z, Yan J, Barriot J-P, Sun S, Haider Z, et al. Evaluating the contribution of Tianwen-4 mission to Jupiter's gravity field estimation using inter-satellite tracking. Astronomy & AstroPhysics, 2025, doi: 10.1051/0004-6361/202554439

In closing this issue, concerning a potential additional comparison with other neural networks, it seems clear that the Reviewer fails to adequately acknowledge that there is not a single definitive number of neural network types, as the field is constantly evolving with new architectures and variations being developed. For the record, here are the main categories:

- Feedforward Neural Networks (FFNNs):

- Perceptron (Single-Layer Perceptron): The simplest form, with only one layer of neurons.

- Multi-Layer Perceptron (MLP): The most common and fundamental type, with at least one hidden layer. It forms the basis for many other architectures.

- Recurrent Neural Networks (RNNs): Designed to handle sequential data, where the output at one step depends on the input and the hidden state of the previous step.

- Long Short-Term Memory (LSTM): An RNN variant that solves the vanishing gradient problem, excellent for long sequences.

- Gated Recurrent Unit (GRU): Similar to LSTMs but simpler, with fewer parameters.

- Bidirectional RNN (Bi-RNN): Processes sequences forward and backward to capture context from both directions.

- Convolutional Neural Networks (CNNs): Excellent for analyzing data with a grid-like topology, such as images (2D) or time series (1D). For example: LeNet, AlexNet, VGG, ResNet, Inception, MobileNet are all xamples of specific CNN architectures that have become standard for image recognition (just our case). Also: 1D, 2D, 3D CNNs depending on the dimensionality of the input data.

- Generative Adversarial Networks (GANs): Composed of two competing networks (a generator and a discriminator) that work together to produce new, realistic data.

- DCGAN (Deep Convolutional GAN), with StyleGAN, BigGAN as more advanced examples.

- Autoencoders: Networks that learn a compressed representation (encoding) of the input data and then attempt to reconstruct the original input (decoding). Used for dimensionality reduction and denoising. Example: Vanilla Autoencoder

- Variational Autoencoder (VAE): Used for data generation.

- Denoising Autoencoder, Sparse Autoencoder: Variants for specific purposes.

- Transformer Networks: Revolutionary for Natural Language Processing (NLP), they rely on a self-attention mechanism to weigh the importance of different parts of the input. For example, BERT, GPT (Generative Pre-trained Transformer) are well-known examples of Transformer-based models.

- Radial Basis Function Networks (RBFNs): Used for function approximation and classification, with neurons in the hidden layer that have radial basis activation functions.

- Self-Organizing Maps (SOMs): A type of unsupervised neural network used for dimensionality reduction and visualizing high-dimensional datasets.

- Liquid State Machines (LSMs) / Echo State Networks (ESNs): A type of recurrent neural network that uses a "reservoir" of neurons with fixed, random connections, where only the output layers are trained.

- Neural Turing Machines (NTMs) / Differentiable Neural Computers (DNCs): Combine neural networks with external, addressable memory, allowing them to learn complex algorithms.

And this list only provides an overview of the main categories, with some of the most prominent architectures. Each category, in turn, includes countless variations and specializations.

What is important is that the choice of a neural network type largely depends on the specific data to be processed and obviously the problem to be solved.

With satellite images, like in our case, the use of a DCNN of MaNet/Imagenet -type is a standard practice and, among experts, should not necessitate additional clarification beyond that already given.

2) Validation test are somewhat limited. A broader range of statistical tests could be applied to test how accurate and reliable the quality of the image identification is. Usually I would expect a wider range, including RMSE and other tests to be there

Answer to 2)

We honestly find it very challenging to believe that requesting RMSE test values is appropriate in this context.

We are sure that the Reviewer knows that Root Mean Squared Error is a common metric typically used in regression problems (which is not exactly our case) to quantify the average magnitude of the errors between predicted values and actual observed values.

In our specific situation (considering that the subject are images), any expert in the scientific world would agree that the measures we have provided - IoU, bIoU, MCC, and then Accuracy and Recall - are far more significant and impactful for validating the results we have obtained than RMSE.

Hence, the decision not to measure RMSE, which would border on the nonsensical, statistically speaking.

3) I still think more can be done with the non-tell based sites but I can accept to leave that for now. Some discussion at least as to how this could be an improvement in the future might be warranted (i.e., how to improve non-tell based site identification as this would really be transformative if we can go beyound mound identification).

Answer to 3)

To respond to this issue, we have inserted the following text in the revised manuscript”:

“While our work has focused on tell-based sites—characterized by distinctive morphologies well-suited to automatic segmentation—it is worth noting that extending this approach to non-mounded contexts would represent a theoretically significant development. However, the lack of recurrent morphological features, the semantic heterogeneity of archaeological traces, and the current scarcity of annotated datasets make such a direction presently difficult to pursue. Advancing in this area would require fundamentally different classification strategies, substantial refinement of source data, and a methodological rethinking that goes beyond the aims and operational scope of the present study.”

---

## [Editor Report · Decision Letter 2]

1 Aug 2025

AI-ming backwards: Vanishing archaeological landscapes in Mesopotamia and automatic detection of sites on CORONA imagery

PONE-D-24-47137R2

Dear Dr. ROCCETTI,

We’re pleased to inform you that your manuscript has been judged scientifically suitable for publication and will be formally accepted for publication once it meets all outstanding technical requirements.

Kind regards,

Shai Gordin, Ph.D.

Academic Editor

PLOS ONE
---

## [Editor Report · Acceptance letter]

PONE-D-24-47137R2

PLOS ONE

Dear Dr. Roccetti,

I'm pleased to inform you that your manuscript has been deemed suitable for publication in PLOS ONE. Congratulations! Your manuscript is now being handed over to our production team.

Kind regards,

on behalf of

Dr. Shai Gordin

Academic Editor

PLOS ONE